# Nexus of Digital Organizational Culture, Capabilities, Organizational Readiness, and Innovation: Investigation of SMEs Operating in the Digital Economy

**Zhang Zhen [1], Zahid Yousaf [2],\*, Magdalena Radulescu [3],\* and Muhammad Yasir [2]**

[1] School of Public Administration, Xi'an University of Architecture and Technology, Xi'an 710000, China; zzhen.china2020@gmail.com

[2] Higher Education Department, Government College of Management Sciences, Mansehra 23100, Pakistan; gccm81@gmail.com

[3] Department of Finance, Accounting and Economics, University of Pitesti, 110040 Pitesti, Romania

\* Correspondence: muhammadzahid.yusuf@gmail.com (Z.Y.); magdalena.radulescu@upit.ro (M.R.)

**Abstract:** This current study was conducted in order to develop a digital innovation model based on the digital organizational culture, digital capability, and organizational readiness. This study explores how the organizational readiness plays a mediating role between the digital capabilities and digital innovation and between the digital organizational culture and digital innovation. For data collection, the survey instrument was used to collect data from 227 SMEs of ICT industry in Pakistan. The findings have revealed a significant connection of the digital organizational culture and digital capabilities with the digital innovation. Second, the organizational readiness mediates between the digital capabilities, digital organizational culture, and digital innovation. The study has empirically confirmed how to flourish a mechanism of the digital innovation in the SMEs. Moreover, the findings suggest some substantial implications for the management by focusing on the digital capabilities and digital organizational culture as a fundamental predictor for the digital innovation via organizational readiness.

**Keywords:** digital capabilities; digital organizational culture; organizational readiness; digital innovation; SMEs; ICT industry

## 1. Introduction

In the current decade, the notion of digital innovation has largely been acknowledged by the researchers to create competitiveness for the all kinds of businesses [1]. Businesses majorly depend on the persistent and influential role of digital innovations to achieve superior performance [2]. The term digital innovation is often recognized as a creation of new ways for doing economic activities using Internet and digital technologies [3]. Internet and the digital technologies consist of Big Data, Internet of Things (IoT), Cloud Computing, augmented and virtual reality, artificial intelligence (AI), and cyber physical systems. Moreover, the digital innovation represents an emerging phenomenon of increasing importance for the SMEs of ICT businesses [1]. The substantial use of information technology in business processes, strategy, marketing, and product development has significantly contributed to modify the businesses' patterns of various industries, including the SMEs of ICT industry [4,5]. Literature shows that only a few studies have been conducted in the ICT and SMEs in order to explore the preconditions for the development of digital innovation [6]. Moreover, researchers overlooked some important factors that accelerate the process of digital innovation within the SMEs of ICT industry. The achievement of the digital innovation for the SMEs of ICT industry is tricky and a complex task and there is a huge need to establish digital organizational culture and digital capabilities. But existing studies scarcely explained the mechanism of improved digital innovation in the SMEs of

ICT industry. Therefore, the current research empirically tests the influence of the digital organizational cultural and the digital capabilities on the digital innovation in the context of the digital economy. The digital economy refers to all the economic activities that are based on ICT, digitized knowledge, and Internet-related information. The effects of the digital economy on the business organizations are numerous and this shift of focus is more profound in the SMEs of ICT industry.

After the spread of Internet and digital technologies in the 21st century the nature of the business concerns around the globe has changed [7]. Delivery of services by using digital technologies demands digital organizational culture and capabilities for the implementation of new business models [8,9]. Digital organizational culture, as defined by some authors [10], represents the set of shared assumptions and overall understanding about the organizational practices in a digital context. The digital organizational culture has become an integral part for the new business model that has set its imprints on the organizational readiness and digital innovation [11]. In line with emerging technologies, the organizations should develop the workplace culture in order to respond to the emergence of advanced technologies [12]. Given the important role of the digital organizational culture for the adaption of the emerging technologies, the organizations capability can manage the best possible use of the technology resources for the innovation process. The organizational digital capability speeds-up the process of innovation by integrating and mobilizing the technologies and human resources [13]. Some authors suggested that the organizations face serious problems for managing digital transformation without preparedness of capability and resources [14]. The digital capabilities of an organization ensure the transformation and integration of digital technologies necessary for readiness and innovation process [9].

Thus, the organizational digital culture and capability play a critical role for the organizational readiness toward a new business model [15,16]. There is no doubt that these constructs gain importance for the direction of the digital innovation. However, it is not enough to improve the digital innovation directly with the organizational digital culture and capability. The existing studies suggested various factors that are the outcomes of the organizational digital culture and capabilities, and at the same time, they positively influence the digital innovation. One of the major outcomes of both digital culture and organizational capabilities is the organizational readiness that consecutively affects the digital innovation [17,18]. Therefore, in this study we highlight this gap and explore the mediating role of organizational readiness between the organizational digital culture, digital capabilities, and digital innovation link. The term organizational readiness represents a firm ability to adjust their available resources for the effective adoption, exploitation, and assimilation of the digital technologies, which supports the implementation of the innovative activities [19]. In the current study, we posit that both organizational digital culture and digital capabilities support the adjustment of the organizational resources with the emerging technologies, which ultimately determine the enhancement of the digital innovation. In the current study we hypothesize the mediating effect of the organizational readiness between the digital organizational culture and digital innovation, as well as between the digital capabilities and digital innovation.

This study aims to identify the connection between the organizational digital culture, digital capabilities, and digital innovation. This study also investigates the meditating role of the organizational readiness. The arrangement of the paper is made in the following pattern. Section 2 presents the literature findings about the organizational digital culture, digital capabilities and digital innovation. Section 3 depicts the methodology of the study. Next section presents data analysis in detail and Section 5 consists of discussions, implications, and conclusion having limitations and directions for future research.

## 2. Literature Review

### 2.1. Digital Organizational Culture and Digital Innovation

Some studies developed the concept of organizational culture, which has become the attention of practitioners and management researchers [20]. It is defined as the mecha-

nism of the way people think, which has a direct influence on the way they behave [21]. Organizational culture is also defined as the beliefs, assumptions and the complex set of values via an organization conducts its business activities [22,23]. Due to the emergence of the technologies and massive online business activities around the world, the culture of the organizations expands in order to include its digital workplace practices [24]. In this sense, the term of digital organizational culture is conceptualized as the shared beliefs, understanding and set of values about the organization of the business activities in a digital domain [10]. The digital organizational culture becomes an integrated part of the organizational life in the digital context [25,26]. Recently, the issues regarding digital organizational culture have been discussed and identified as the integrated part of the organization in the era of the advanced technologies [27]. On the other hand, digital innovation also becomes an emerging construct for the organization because of the online economic activities [28]. The concept of the digital innovation has tight connections with the concept of creativity, meaning that it settles for the organizations a creative behavior and implementation of some new methods for solving the existing business problems by using the new emerging technologies [3,6]. In the recent decade the concept of innovation has been changed or expanded into digital innovation [4]. Digital innovation is defined as the application of the advanced digital mechanisms and technologies for solving or improving the existing business processes and for launching some advanced business models [3].

The debate on the link between the digital organizational culture and digital innovation mentioned in the previous studies [8,27] has found that digital organizational culture is a significant predictor for the application of the digital technologies which provides new methods and solutions for business process through advanced business models. Some authors have also argued that digital organizational culture positively affects the organizational digital innovation [27]. Studies have also stated that organizations having a digital organization culture are more likely to show higher digital innovation [13]. Based on these arguments, we hypothesized that:

**Hypothesis 1 (H1).** *Digital organizational culture has positive relationship with the digital innovation.*

### 2.2. Digital Capability and Digital Innovation

Digital capability is referred to as the mechanism that ensures the transformation and integration of technological resources and make the best use of these technological resources [29]. A digital capability which is considered as an integrated part of the digital innovation is crucial as it provides integration of the digital technologies with the digital professionals [30]. Digital capabilities play an important role for the improvement of the digital innovation [20]. Based on the theory of dynamic capability, the digital capability is considered as an organization's dynamic capability which describes the firm's ability to produce new products and business model in order to respond to the dynamic circumstances [31]. When organizations are more inclined toward digital capabilities in term of their ability to create new methods for the business processes, this increases the level of the digital innovation at the organizational level [32]. Some previous studies have shown the link between the digital capabilities of an organization and the digital innovation [33,34]. According to some authors, the digital capabilities represent a significant predictor for the organization's digital innovation [30]. Some studies have also stated that the organization's ability to integrate the human and technological resources in response of the dynamic environment gives more confidence for creating and implementing of new business models [20]. Based on these arguments, we hypothesized that:

**Hypothesis 2 (H2).** *Digital capabilities have positive relationship with the digital innovation.*

### 2.3. Digital Organizational Culture and Organizational Readiness

Organizational ability to implement and utilize the existing resources for the improvements of the business process in the setting of the virtual environment named as

organizational readiness has gained more importance for the implementation of the technological improvements [21,35]. Organizational readiness is considered as the main driving force for reacting in an unpredictable environment [36]. Organizational ability to adopt and implement the technological improvements becomes necessary because of the transformation of the traditional business activities into the digital domain [37]. Excessive development of the technological resources demands better use of the existing human and technologies resources for the improvement of the business activities [38]. Organization culture that supports the digital transformation enhances the organizational ability to utilize its resources for the better functioning of the business activities [21]. According to some authors, the digital organizational culture is a significant predictor of the organizational readiness toward the implementation of the advanced business models with the help of the digital technologies [10]. Moreover, some researchers found that the digital organizational culture which meets the demands of the digitalization has a positive effect on the organizational readiness in the virtual environment [8]. In line with these studies, we hypothesized that:

**Hypothesis 3 (H3).** *Digital organizational culture has positive relationship with the organizational readiness.*

### 2.4. Digital Capabilities and Organizational Readiness

According to the dynamic capability theory, the digital capability of an organization is considered as a dynamic capability which describes its ability to create a new business process in response of the technological improvements in a turbulent business environment [39]. The organizational capability to produce new products and business models increases the organizational ability to utilize the resources for the improvements of the business activities in the setting of the virtual environment [37]. Organizational capability has gained more importance and become the main driving force in the emerging technological business environment [30].

Moreover, the digital capabilities of an organization support the digital alignment because only those organization having skills and capabilities can control and manage new technologies in order to convert these advanced technologies into new business products [40]. The organizational digital capabilities boost commitment and readiness to adopt and implement technological improvements [41]. The digital capabilities enhance the organizational readiness which, in turn, increases the firm's ability to utilize its resources in some better ways [39]. According to some authors, the digital capabilities represent a significant predictor of the organizational readiness toward the adoption and implementation of the advanced business models [29]. Moreover, some studies have found that digital capabilities, which represent the dynamic capability of an organization in the era of digitalization, have positive effect on the organizational readiness [17]. In line with these studies, we hypothesized that:

**Hypothesis 4 (H4).** *Digital capabilities have positive relationship with the organizational readiness.*

### 2.5. Organizational Readiness and Digital Innovation

Digital innovation represents the mechanism that involves the organization creative behavior for the implementation of new methods in order to solve the existing problems via the emerging technologies [2]. An organization which is more inclined toward the application of the advanced technologies for solving the business problems, presents better ability and skills to integrate and mobilize the existing resources for generating new business process [21]. The organization's confidence and credibility regarding its ability for the best utilization of the existing human and technological resources improve the level of the digital innovation [42]. This ability of the firm increases the level of integration of the existing resources which in turn improves the organizational digital innovation [43]. Higher level of readiness enhances the organization commitment which

can determine the best possible use of the resources and this is considered as a sign of digital innovation [44]. When organizations improve the readiness level of their human and technology resources, this enhances the level of the digital innovation [45]. Moreover, it is argued that a higher level of the organizational readiness enhances the level of the organizational digital innovation [19].

**Hypothesis 5 (H5).** *Organizational readiness has positive relationship with the digital innovation.*

### 2.6. Mediating Role of Organizational Readiness

The digital innovation becomes an emerging phenomenon for the business organization [1]. The reasons behind this emerging phenomenon are due to the involvement of the digital technologies in the business activities [8]. The organizations are continuously trying to implement and utilize the existing human and technological resources for the necessary improvement of their business activities. Firm's ability to integrate and utilize the existing resources positively affects the organizational digital innovation [43]. Therefore, organizational readiness is a valuable mean to achieve the organizational digital innovation [30]. By this perspective, organizational readiness plays an important role and acts as a bridge between the digital organizational culture and digital innovation. The mediating role of the organizational readiness between the digital organizational culture and digital innovation can be explained and justified with the help of this logic. The digital organizational culture provides infrastructure, shared assumptions, and a set of values that are the foundation for the digital innovation. Therefore, it is argued that the organizational digital culture ensures the development of the shared assumption and set of standard values, which ultimately enhances the digital innovation. In line with the above assumptions, we formulated the hypothesis:

**Hypothesis 6 (H6).** *Organizational readiness positively mediates the relationship between the digital organizational culture and digital innovation.*

As it has been already discussed previously, the digital capabilities positively predict the organizational readiness [20]; moreover, the organizational readiness positively affects digital innovation [19]. Therefore, based on the above arguments, we argue that digital capabilities affect the digital innovation through the organizational readiness. Digital capabilities are the dynamic capabilities of an organization that increase the ability to create new business models in response to the emerging technologies [17], which in turn enhances the digital innovation. Digital capability enhances the learning mechanism and also enables the organizations to quickly understand the new technologies. These newly learned techniques through improved digital technology crop-up the organizational ability of adopting the organizational readiness in order to accept the change. Once the organizations have the ability to accept newest technologies smoked by digitalization, those organizations can attain digital innovation very easily. Hence, we build the last hypothesis of the study that digital capability affects digital innovation through organizational readiness.

**Hypothesis 7 (H7).** *The relationship between the digital capabilities and digital innovation is mediated by the organizational readiness.*

### 2.7. Theoretical Framework

This study used four constructs i.e., digital organizational culture and digital capability (independent), organizational readiness (mediators) and digital innovation (dependent). The details are shown in Figure 1.

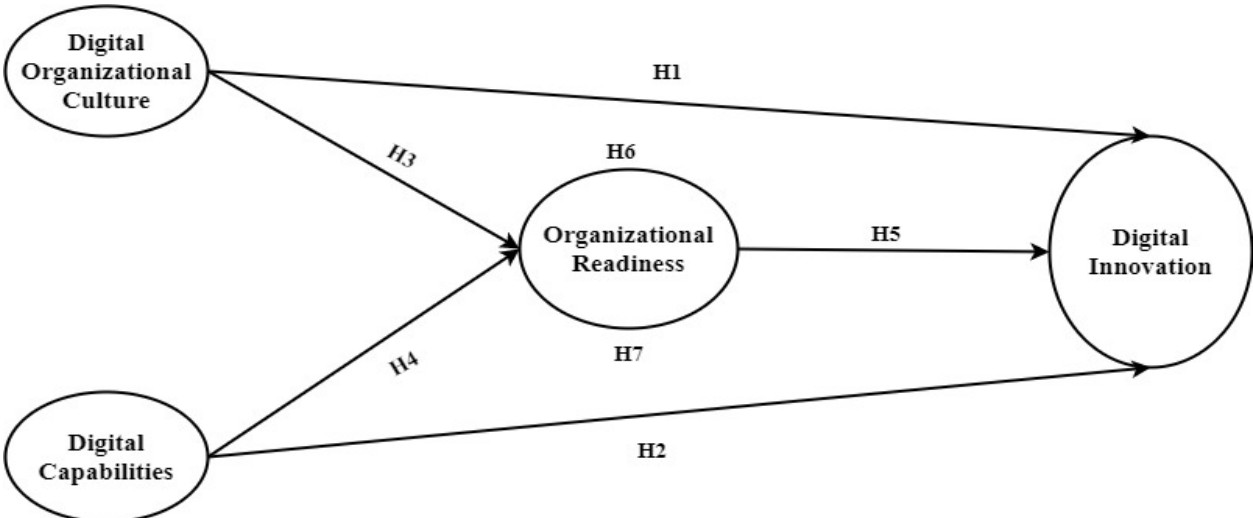

**Figure 1.** Theoretical framework.

## 3. Methodology

### 3.1. Data Collection and Participants

The current study has utilized a cross-sectional research design for the investigation of the hypotheses we have elaborated above. The targeted sample has included the owner/managers of the SMEs in ICT industry in Pakistan. Information about the SMEs was taken from the databases of SMEDA (small medium enterprises development authority) and Chamber of Commerce and Small Industrial Development Board (SDIB). As per SMEDA, the SMEs are the business enterprises having more than 10 employees and less than 250 employees. Questionnaires were distributed to 413 owner/managers of the SMEs. Only 227 responses were collected during the data collection process. Data for this study were collected in two phases. The reasons for conducting the survey in two different phases were to reduce respondent fatigue and to minimize the chances of common method variance by applying temporal separation [46]. Before launching full range data collection process, a pilot study was carried out on 25 managerial officials from the targeted SMEs for assuring the clarity, suitability, as well as relevancy of the research instrument. With a view to ensure the proper understanding of the nature of this study vis-a-vis pertinent constructs, the questionnaire addressed to the managers was provided in the local language (i.e., Urdu) and later on the same was transcribed back into English version. The questionnaire was checked from the experts and academia to avoid any confusion or deficiency, before its distribution among the respondents.

### 3.2. Measurement

Respondents of the current study provided information through structured questionnaire pertaining to digital organizational culture, digital capabilities, organizational readiness, and digital innovation (see Appendix A). Measurement scale consists of five-points i.e., 1 to 5.

#### 3.2.1. Digital Organizational Culture

Digital organizational culture was measured with four-items scale formulated and used by other authors [47] and [48] (see Appendix A). Sample item include "The organization shares with the staff the digital strategy, taking into consideration their suggestions." The current study reported alpha value of 0.86 for these four items (see Table 1).

**Table 1.** Composite reliability, and average variance extracted.

| Constructs | Factor Loading | AVE | CR | Cronbach's α |
|---|---|---|---|---|
| Digital Innovation | | | | |
| DI1 | 0.882 | | | |
| DI2 | 0.859 | | | |
| DI3 | 0.824 | 0.70 | 0.93 | 0.82 |
| DI4 | 0.773 | | | |
| DI5 | 0.843 | | | |
| DI6 | 0.833 | | | |
| Digital capabilities | | | | |
| DC1 | 0.883 | | | |
| DC2 | 0.866 | | | |
| DC3 | 0.785 | 0.71 | 0.92 | 0.84 |
| DC4 | 0.859 | | | |
| DC5 | 0.828 | | | |
| Digital organizational culture | | | | |
| DOC1 | 0.893 | | | |
| DOC2 | 0.886 | 0.75 | 0.92 | 0.86 |
| DOC3 | 0.789 | | | |
| DOC4 | 0.891 | | | |
| Organizational readiness | | | | |
| OR1 | 0.872 | | | |
| OR2 | 0.883 | | | |
| OR3 | 0.785 | 0.75 | 0.94 | 0.83 |
| OR4 | 0.869 | | | |
| OR5 | 0.891 | | | |
| OR6 | 0.863 | | | |

Note: CR: composite reliability; AVE: average variance extracted.

### 3.2.2. Digital Capabilities

Digital capabilities were measured with five-items scale formulated and used by previous studies [49] (see Appendix A). Sample items include "identifying new digital opportunities." The current study reported alpha value of 0.84 for these five items (see Table 1).

### 3.2.3. Organizational Readiness

Organizational readiness was measured with the help of six-item scale formulated and used by some other authors [50] (see Appendix A). Sample items include "when changes are necessary, management provides a clear plan for implementing." The current study reported alpha value of 0.83 for these six items (see Table 1).

### 3.2.4. Digital Innovation

Digital innovation was measured with the help of six-item scale formulated and used by some previous studies [49] (see Appendix A). Sample items include "the quality of our digital solutions is superior compared to our competitors". The current study reported alpha value of 0.82 for these six items (see Table 1).

## 4. Results and Analysis

For the purpose of the data analysis, the current study has used the descriptive statistics, correlation, multiple hierarchical regressions techniques, and structural equation modelling. Moreover, the discriminant validity was examined by using AMOS 7.0 software with the help of the confirmatory factor analysis (CFA). The aim of the current study was to find out the liner relationships between variables used in the hypothesized model. In order to examine the liner relationships, we employed the covariance approach. The covariance-based approach measures how two random variables change at the same time.

On the other hand, the variance-based approach tells us how the data set is spread around its mean value, while the covariance-based approach is used to determine the directional relationship between the analyzed variables.

The SEM technique was used for the confirmatory factor analysis (CFA). The discriminant validity of the study constructs was established with the help of the CFA techniques recommended by other authors [51]. In the current study, four models were verified with different configurations. Model-1 contains the organizational digital culture, digital capabilities, organizational readiness, and digital innovation into one factor. Model-2 contains two-factor, organizational digital culture, digital capabilities, and the organizational readiness merged into first factor, while digital innovation into second factor. Third model structure consists of three-factor, the organizational digital culture and digital capabilities merged into one factor and the organizational readiness is the second factor, while the digital innovation is the third factor. Model-3 contains the organizational digital culture, digital capabilities, organizational readiness, and digital innovation. The outcomes of the CFA analysis have revealed that the data were good fit for the four-factor hypothesized model i.e., ($\chi^2$ = 611.13; $df$ = 984; $p < 0.001$; CFI = 0.92; GFI = 0.94 and SRMR = 0.057).

### 4.1. Descriptive Analysis

The values of correlations, mean, and standard deviation (SD) are presented in Table 2. Table 2 shows that there is a significant and positive connection between all constructs including: independent, mediator, and dependent. Table 2 presents a positive relationship between the digital organizational culture and the organizational readiness (0.33**), digital innovation (0.21**), digital capabilities, and organizational readiness (0.41**), digital innovation (0.19**), organizational readiness, and digital innovation (0.39**).

**Table 2.** Correlation.

| Variables | Mean | SD | 1 | 2 | 3 | 4 | 5 |
|---|---|---|---|---|---|---|---|
| Business Age | 3.2 | 0.79 | 1 | | | | |
| Business Size | 0.3 | 0.86 | 0.09 | 1 | | | |
| Digital org. culture | 2.7 | 0.82 | 0.06 | 0.01 | 1 | | |
| Digital capabilities | 2.2 | 0.89 | 0.04 | 0.03 | 0.24 ** | 1 | |
| Organizational readiness | 3.7 | 0.91 | 0.07 | 0.10 * | 0.33 ** | 0.41 ** | 1 |
| Digital innovation | 3.5 | 0.89 | 0.03 | 0.09 | 0.21 ** | 0.19 ** | 0.39 ** |

Note: (* $p < 0.05$, tow tailed) (** $p < 0.01$, two tailed).

### 4.2. Testing Hypotheses

The above-mentioned hypotheses were tested with the help of hierarchical regression analysis. Table 3 and Figure 2 depict the outcomes of the regression analysis. Results presented in Model 1, 2, 3, and 4 from Table 3 explain the direct effect among construct of the study. Regarding hypothesis 1 i.e., digital organizational culture significantly predicts digital innovation. Model 3 contains the coefficients of the direct effect of the digital organizational culture and digital innovation (ß = 0.21**). The coefficients depicted in Model 3 from Table 3 confirmed the Hypothesis 1.

For supporting hypothesis 2, Model 3 from Table 4 also provides the findings of the direct effect of the digital capabilities and digital innovation. Digital capabilities positively predict the digital innovation (ß = 0.19**). The coefficients depicted in Model 3 of Table 4 confirm the Hypothesis 2 of this study.

Furthermore, for supporting Hypothesis 3, Model 2 from Table 3 provides the findings of the direct effect of the digital organizational culture on the organizational readiness. The results confirm that the digital organizational culture is positively predicting the organizational readiness (ß = 0.33**). The coefficients depicted in Model 2 from Table 3 confirm the Hypothesis 3 of this study. Moreover, regarding study Hypothesis 4, the direct effect of the digital capabilities on the organizational readiness is also confirmed, as the

regression coefficients presented in Model 2 from Table 4 show a positive direct effect of the digital capabilities on the organizational readiness ($\beta = 0.41$**).

**Table 3.** Hierarchical regression results.

| Variable | Organizational Readiness | | | Digital Innovation | |
| --- | --- | --- | --- | --- | --- |
| | Model 1 | Model 2 | Model 3 | Model 4 | Model 5 |
| Business Age | 0.011 | 0.01 | 0.006 | 0.007 | 0.011 |
| Business Size | 0.083 | 0.08 | 0.081 | 0.029 | 0.089 |
| Digital Org. Culture | | 0.33 ** | 0.21 ** | | 0.14 * |
| Organizational Readiness | | | | 0.39 ** | 0.42 ** |
| $R^2$ | 0.032 | 0.36 | 0.38 | 0.32 | 0.38 |
| $\Delta R^2$ | 0.031 | 0.33 | 0.35 | 0.31 | 0.34 |
| F | 4.17 * | 35.4 * | 40.98 * | 33.06 ** | 20.98 * |
| $\Delta F$ | 4.17 * | 17.07 * | 16.09 * | 19.7 ** | 13.01 * |

Note: * $p < 0.05$; ** $p < 0.01$.

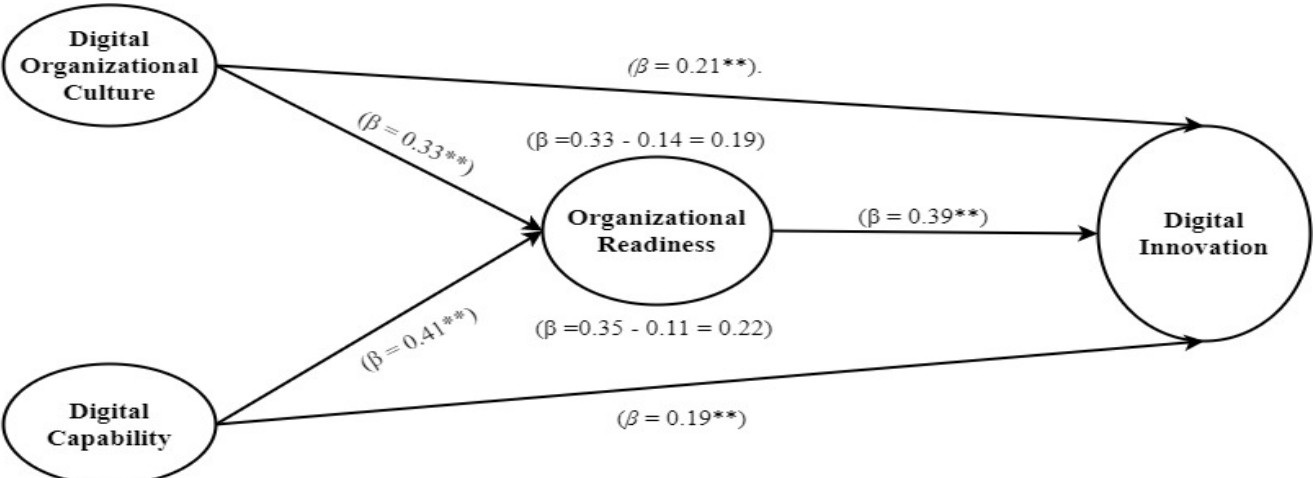

**Figure 2.** Coefficients of regression analysis. ** $p < 0.01$, two tailed.

**Table 4.** Hierarchical regression results.

| Variable | Organizational Readiness | | | Digital Innovation | |
| --- | --- | --- | --- | --- | --- |
| | Model 1 | Model 2 | Model 3 | Model 4 | Model 5 |
| Business Age | 0.012 | 0.009 | 0.005 | 0.008 | 0.014 |
| Business Size | 0.08 | 0.081 | 0.08 | 0.025 | 0.086 |
| Digital Capabilities | | 0.41 ** | 0.19 ** | | 0.11 * |
| Organizational Readiness | | | | 0.39 ** | 0.42 ** |
| $R^2$ | 0.032 | 0.36 | 0.38 | 0.32 | 0.38 |
| $\Delta R^2$ | 0.031 | 0.33 | 0.35 | 0.31 | 0.34 |
| F | 4.17 * | 35.4 * | 40.98 * | 33.06 ** | 20.98 * |
| $\Delta F$ | 4.17 * | 17.07 * | 16.09 * | 19.7 ** | 13.01 * |

Note: * $p < 0.05$; ** $p < 0.01$.

Model 4 from Table 3 provides the findings for the direct effect of the organizational readiness on the digital innovation. The results also confirm that the organizational readiness significantly predicts the digital innovation ($\beta = 0.39$**). The coefficients depicted in Model 4 confirm the Hypothesis 3.

Results presented in Model 5 from Tables 3 and 4 explain the mediating role of the organizational readiness. Model 5 from Table 3 shows that the organizational readiness mediates between the digital organizational culture and the digital innovation. After the

organizational readiness is added, the coefficient of the digital organizational culture is reduced from β = 0.21** to β = 0.14*), while the coefficient of the organizational readiness is (β = 0.42**). These findings suggested that organizational readiness partially mediates between the digital organizational culture and the digital innovation. Thus, Hypothesis 6 is supported.

Model 5 from Table 4 also presents the findings for the mediating role of the organizational readiness in explaining the association between the digital capabilities and the digital innovation. After organizational readiness is added, the coefficient of digital capabilities is reduced from (β = 0.19** to β = 0.11*), while the coefficient of the organizational readiness is (β = 0.42**). These findings suggested that the organizational readiness partially mediates between the digital capabilities and the digital innovation. Thus, Hypothesis 7 is supported. The results of the normal test theory approach elaborated by some other authors [52] are presented in Table 5. They also verify the indirect effects of the digital organizational culture on the digital innovation. Table 5 shows the total effect (β = 0.33**, t = 6.52), the direct effect (β = 0.14**, t = 1.89) and the indirect effect (β =0.33 − 0.14 = 0.19) of the digital organizational culture on the digital innovation through the organizational readiness. Sobel test i.e., (Z = 3.52) supports H6 that the digital organizational culture affects the digital innovation via organizational readiness. Hence the hypothesis 6 is supported.

**Table 5.** Direct and indirect effects of digital organizational culture and digital capabilities on digital innovation using the SPSS version of PROCESS.

| Mediation Models | Total Effect | | | Direct Effect | | | Indirect Effect | | |
|---|---|---|---|---|---|---|---|---|---|
| | | | | | | | Normal Test Theory | | |
| | β | t | p | β | t | p | β | Z | p |
| DOC→ OR→ DI | 0.33 | 6.52 | 0 | 0.14 | 1.89 | 0.09 | 0.19 | 3.52 | 0 |
| DC→ OR→ DI | 0.35 | 8.72 | 0 | 0.11 | 1.39 | 0.09 | 0.24 | 3.72 | 0 |

Note: DOC (digital organizational culture); OR (organizational readiness); DC (digital capabilities); DI (digital innovation).

Table 5 also enlists the outcome of the indirect effect of the digital capabilities on the digital innovation. Table 5 shows the total effect (β = 0.35**, t = 8.72), the direct effect (β = 0.11**, t = 1.39), and the indirect effect (β =0.35 − 0.11 = 0.22) of the digital capabilities on the digital innovation through organizational readiness. Sobel test i.e., (Z = 3.72) proves that the digital capabilities affects the digital innovation via organizational readiness. Hence the study hypothesis 7 is supported. Figure 2 shows the results of all hypotheses tested in the current research.

## 5. Discussions of the Results

This study affirms seven hypotheses to explore the impact of the digital organizational culture, digital capabilities, and organizational readiness on the digital innovation. H1 shows that the digital organizational culture predicts the digital innovation. The results prove that the digital innovation mainly depends on the digital organizational culture. SMEs of ICT industry can ensure the digital innovations through the development of the digital organizational culture i.e., embracing the digital transformations and extensive use of the digital technologies, artificial intelligence, big data analytics, IoT platforms, multilevel customer interaction, and cloud computing. This is the first study that offers unique findings and extends the bright harmony of the digital organizational culture beyond its simple link with the performance or competitiveness of the organizations. The findings about the H1 reveal that the outcomes of the digital organizational culture are not only meagerly attached to the innovation [8], transformation [13], but also to the digital innovation [27]. Hence, the results of H1 clearly show the novelty, significance, and the difference against the findings of the previous studies and show that SMEs of ICT industry

can achieve the digital innovation through the digital innovation culture in the digital economy context [8,13,27].

H2 shows the direct link between the digital capabilities and digital innovation. The results of H2 confirm the positive association between the digital capabilities and digital innovation. These results demonstrate that the digital innovation in the SMEs of ICT industries largely depends on the integration and mobilizing of the technological resources. Existing empirical studies also found a positive association between digital capabilities and digital innovation [20,30]. Therefore, H2 is confirmed.

H3 of the study shows that digital organizational culture is positively associated with the organizational readiness. These results confirm that the digital organizational culture positively contributes toward the organizational readiness through the implementation and utilization of the existing resources of the firm for the improvement of the business processes. These findings confirm the H3 of the study and they are consistent with the findings of the previous studies [8,10]. Furthermore, H4 of the study shows a positive relationship between the digital capabilities and the organizational readiness. The results reveal that the association between the digital capabilities and the organizational readiness is also statistically supported and confirm H4. The previous studies [17,29,39] also suggested a positive association between the digital capabilities and the organizational readiness. H5 of the study confirms a positive relationship between the organizational readiness and the digital innovation. The findings about the H5 reveal that the organizational readiness significantly contributes to the digital innovation. Sixth, the mediating role of the organizational readiness in the relationship between the digital organizational culture and the digital innovation is also confirmed. These findings reveal that digital organizational culture contributes to the improvement of digital innovation in the presence of the organizational readiness. Hence, the H6 of this study is supported. Finally, this study confirms the mediation effect of the organizational readiness between the digital capabilities and the digital innovation according to the result that confirm H7.

## 6. Conclusions

The current study considered the mechanism of the digital innovation of the SMEs based on the digital culture and capabilities of these SMEs of ICT industry in Pakistan. Moreover, the mediating role of the organizational readiness has also been tested. Seven hypotheses were formulated and validated in the current study. The findings have shown that the digital organizational culture has a positive relationship with the digital innovation of the SMEs firms. Furthermore, the results also confirmed that the digital capabilities also predicted the digital innovation in a positive way. Moreover, the digital organizational culture and digital capabilities have a positive association with the readiness of the SMEs of ICT industries. Finally, the mediating role of the organizational readiness has also been confirmed by the findings of this study.

### 6.1. Contributions to Theory

This research contributes to theory especially in the domain of the digital innovation, digital organizational culture and capabilities in determining the digital innovation. There is a limited number of studies that explain the mechanism of how the digital organizational culture and digital capabilities enhances the digital innovation. This research explores the mechanism of the digital innovation improvement of the SMEs through the digital culture and capabilities. Second, the current study develops a digital innovation model for the SMEs which explains how a combination of different factors like the digital organizational culture, digital capabilities, and the organizational readiness can influence the digital innovation of the SMEs. These findings are consistent with the existing studies that empirically validated the relationship between the digital organizational culture understanding and a set of values about the organization as well as the shared beliefs and digital innovation [13,27]. Furthermore, the transformation, integration, and best use of the technological

resources play an important role for the enhancement of the digital innovation within an organization [30,33].

The third contribution of this research refers to the examination of the digital culture and capabilities for the improvement of the organizational readiness. The organizational readiness represents an important ability of an organization that ensures the participation into innovation-related activities [49]. The current study considers this gap and focuses on the digital culture and capabilities as a pre-condition for the organizational readiness and digital innovation.

Fourth, although the existing literature has already explained that the organizational supportive culture and the dynamic capabilities is the key to the organizational innovation process, however, the researchers have given limited attention to the effects of the digital capabilities on the digital innovation [30,31]. Only a few studies have suggested that the digital organizational culture and digital capabilities provide a strong base for innovation through the organizational readiness. In this study, we have investigated and found support for the mediating effect of the organizational readiness on the digital innovation. Very few studies documented, so far, any mediating mechanism by which organizational abilities such as integration and mobilization of the existing resources predict the digital innovation. Furthermore, few scholars have examined the role of the organizational readiness, which is a powerful indicator behind the digital innovation [19]. Being an outcome of the digital culture and digital capabilities, the organizational readiness has a major impact on the digital innovation. The current results confirm that the organizational readiness mediates between the digital organizational culture, digital capabilities, and digital innovation. The aim of this study is not only to explain the effect of the digital organizational culture and digital capabilities on the digital innovation, but also to contribute to accelerating the organizational readiness and digital innovation with the support of the digital organizational culture and digital capabilities.

### 6.2. Practical Implications

This research has some significant implications for the management in practice. First, our findings suggest that the digital organizational culture and digital capabilities can enhance innovation activities within an organization with the help of the organizational readiness, by focusing on the integration and mobilization of the existing human and technological resources [43,53]. By doing so, the participation in the innovation-related activities can only be possible when organizations have the ability to integrate and mobilize the firm's resources. Developing of the digital organizational culture and digital capabilities is not sufficient for achieving the digital innovation; the organizational readiness is equally important for the execution of the innovation-related activities [30]. Second, this study suggests that the digital organizational culture and digital capabilities are the pre-conditions and powerful predictor of the organizational readiness. Therefore, in order to increase the level of the organizational readiness and digital innovation, the organizations must focus on the development of the digital culture and improve the digital capabilities that will help them enhance their ability toward the digital innovation [54,55].

### 6.3. Limitation and Future Research

Besides the various theoretical and practical implications, some limitations were also noticed that might be important for future research. First, the current study focused on organizational readiness as a mediating variable between digital organizational culture and digital innovation; some other constructs such as organizational flexibility and organizational ambidexterity can also be tested and validated as a mediating variable in the exiting links of the current study. Second, the current study only considered the SMEs, therefore its findings cannot be generalized to the other sectors. Moreover, due to the context differences, the findings of the current study cannot be generalized to other SMEs of other countries. Finally, some other exogenous variables may affect the association we have studied. In future, the researchers are required to focus on some other important

determinants of the digital innovation like bundle of HR practices, high involvement in HR practices, and high-performance work system etc. Therefore, the future research should be conducted by considering some other factors as well. It would be also interesting to analyze the relation between our two exogenous variables, the digital organizational culture and digital capabilities, which could lead to some interesting results.

**Author Contributions:** Conceptualization, and methodology: Z.Z., Z.Y.; software and validation: M.Y.; formal analysis: M.Y.; investigation: Z.Z.; resources: Z.Z.; data curation: M.Y.; writing—original draft preparation: Z.Y.; writing—review and editing: M.R.; visualization: M.R. and Z.Y.; supervision: Z.Y.; project administration: Z.Y. and Z.Z. All authors have read and agreed to the published version of the manuscript.

**Funding:** Supported by: Foundation for Teachers, Xi'an University of Architecture and Technology, China (RC19008); Educational Commission of Shaanxi Province, China (20JK0214); Social Science Foundation of Shaanxi Province, China (2020D014).

**Institutional Review Board Statement:** Both national and international standard were considered for the completion of current research, also approval was obtained from SMEDA (small & medium size enterprises) for the collection of data from owner/managers of SMEs. The Institutional Review Board Letter No. GCMS2020/Adm/32. The current study didn't harm any humans or animals.

**Informed Consent Statement:** Consent was obtained from the participants i.e., owner/managers of SMEs of Pakistan.

**Data Availability Statement:** The data that support the findings of this study are not publicly available due to respondents' confidentiality reasons. The data are not publicly available due to containing information that could compromise the privacy of the participants in this research.

**Conflicts of Interest:** The authors declare no conflict of interest.

## Appendix A

**Digital innovation** (Paladino, 2007)
DI1. The quality of our digital solutions is superior compared to our competitors'.
DI2. The features of our digital solutions are superior compared to our competitors'.
DI3. The applications of our digital solutions are totally different from our competitors'.
DI4. Our digital solutions are different from our competitors' in terms of product platform.
DI5. Our new digital solutions are minor improvements of existing products.
DI6. Some of our digital solutions are new to the market at the time of launching.
**Organizational Readiness** (Claiborne et al., 2013)
OR1. We understands that specific changes may improve outcomes.
OR2. Most staffs are willing to try new ideas and it is easy to change procedures to meet new conditions.
OR3. Staff members ask questions and express concerns about changes.
OR4. When changes are necessary, management provides a clear plan for implementing.
OR5. Staff are encouraged to discuss and explore evidence-based practice techniques.
OR6. Staff adapts quickly when they have to shift focus to accommodate program changes.
**Digital capabilities** (Paladino, 2007)
DC1. Acquiring important digital technologies.
DC2. Identifying new digital opportunities.
DC3. Responding to digital transformation.
DC4. Mastering the state-of-the-art digital technologies.
DC5. Developing innovative products/service/process using digital technology.
**Digital organizational culture** (Denison and Mishra, 1995; Büschgens et al., 2013)
DOC1. The teams collaborate functionally in the initiatives for the innovation and digital transformation.
DOC2. There is a clear orientation to digital technology changes inside the organization's culture.
DOC3. The culture of digital innovation and change takes part as a natural process within the organization.

DOC4. The organization shares with the staff the digital strategy, taking into consideration their suggestions.

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
