# Peer review of "Nexus of Digital Organizational Culture, Capabilities, Organizational Readiness, and Innovation: Investigation of SMEs Operating in the Digital Economy"

_sustainability, doi:10.3390/su13020720_

Round 1
Reviewer 1 Report
First of all I would like to say that the paper structure is clear, the research design is proper, however some shortcomings are visible and need to be clarify. Below I put comments with regard to noticed questionable issues which require to be review and refine.
Introduction
1. Please explain what type of technology the Authors mean by digital technology as it covers broad spectrum. From Authors latent explanation it seems that their define it as just ICT.
2. In my opinion the phrase “digital organizational cultural and the digital capabilities on the digital innovation in the context of the digital economy” (also repeated in the paper title) too much emphasizes ‘digital’ causing some of partial expressions not clear. For example, digital organizational culture means that culture is digital; digital economy is used not for describing business ecosystem or national level of economy but just refers to organizations however its definition is broad. Nevertheless business economy is omitted in defining the paper’s objective. All these inconsistencies makes the that paper focus is unevenly described in title, research objective, and research justifications. My recommendation is to review the title to clear it from unnecessary digital inclusions and to harmonize mentioned paper’s parts.
Literature review
1. the definition of digital organization culture is missing. The H1 is not precise: what the Authors mean by “affects digital innovation”? Its creation? Adoption? Diffusion? Organizational innovativeness? Moreover the H1 is defined as causal (affects) why the other as correlation.
2. The definition of digital capability is tautological “defined as the firm ability and *capability*…”
3. The same remark refers to H2 as it is not clear how digital capabilities are correlated.
4. H3 and H4 sound as propositions without referring to correlation but only as predictor which is not clear.
5. The hypotheses should be harmonized as they are applied to the same theoretical framework. In the model the key expressions are ‘effects’. However the framework is built from previous hypotheses and it complete.
6. I wonder if there is one more possible relationship: digital organizational culture and digital capabilities. Could Authors refer to this?
Methodology
1. The sample of 227 is not representative and it hampers all the research. The process of respondents selection should be specified. The questionnaire should be added as appendix. The estimation of sample coverage from ICT industry should be justified. There is lack of data when survey was done and how the answers were collected. This part needs to be substantially improved as in current form its reliability is low.
2. Section 3.2 is current form in not clear. If it supposed to be review of research instrument used then more details are needed such as an exact full questions asked, and values should be in table form – however table with Cronbach’s alfa value is provided later. It is necessary to explain how key constructs like digital innovation were defined in questionnaire – partial codes (latent variables) are not clear and does not allow to validate the proper construction of key variables.
3. As the papers refer to SMEs it is necessary to provide applied definition. Also sample description would be good to described like number of employees (size?), turnout, age, etc.
4. I recommend to add data analysis section where applied methods of analysis like SEM, descriptive statistics, regression analysis were used also explaining steps of analysis.
Results and analysis
1. The explanation of using SEM should be in methodology section as well as mention about software AMOS and SPSS.
2. The calculations are presented in satisfactory manner.
Discussions and Conclusions
1. I propose to put all conclusions on hypotheses verification into one table showing what was verified. Currently there is a description however from previous part of the text we know that in different models hypotheses were positively tested.
2. Reading explanations I got impression that the Authors proved what was already known and it is good that they separate section 5.1. However a “digital innovation Model (sic!) for SMEs” is missing.
3. Please review phrases referring to impact and correlation as correlation does not mean causation.
4. Here the Authors explain (line 407) that digital innovation is “the level of the innovation activities”. The phrasing definitely need to harmonize across whole paper.
5. I see the problem of demonstrating contributions to theory by lack of details. For example, the Authors refers to “mechanism how the digital organizational culture and digital capabilities enhances the digital innovation” however the details of its mechanism are not covered by the papers as there is no explanation of variables. It is not clear what constitutes digital organizational culture or digital capabilities. In my opinion the partial variables should be derived from literature review while the Authors used more general terms and very often ambiguous like digital capabilities and test. Without a good explanation how construct are defined, what are their components, etc. the computation would does not actually confirm what we know.
6. The second contribution is declarative (line 382-383).
7. The section of limitation should be developed and the findings refer to Pakistan SMEs – are they representative for the other countries?
Technical issue: Space between words is missing in line 44, 45, 67, 75, 121, 269 etc.
Good luck with the revision!
Author Response
Dear Reviewer,
Thank you very much for your valuable comments and suggestions for improving our paper! We have tried to address to all your suggestions. We have marked those changes in yellow color. We attach the Response Letter for you with the details of the required changes.
Thank you very much once again!

Reviewer 2 Report
I enjoyed reading this paper, good job.
Here are some comments to strengthen your work.
I think that seven hypotheses are a bit too many, please take it into consideration for the next times... As every hypothesis should lead to a comprehensive discussion. Indeed, the discussion part is too short with respect to the results. Please enlarge the discussion part, and separate the conclusions.
The methodological part needs some references for the methodology employed. Please refer to previous studies employing the same methodology and cite them.
Moreover, I strongly suggest adding more references.
Please consider including these works:
Bagnoli, C.; Dal Mas, F.; Massaro, M. (2019) The 4th Industrial Revolution and its features. Possible business models and evidence from the field, International Journal of E-Services and Mobile Applications, vol. 11, issue 3, article 3, pp. 34-47
Cohen, B., Amorós, J.E. and Lundyd, L. (2017), “The generative potential of emerging technology to support startups and new ecosystems”, Business Horizons, Vol. 60 No. 6, pp. 741–745.
Elia, G., Margherita, A. and Passiante, G. (2020), “Digital entrepreneurship ecosystem: How digital technologies and collective intelligence are reshaping the entrepreneurial process.”, Technological Forecasting and Social Change, Vol. 150, p. 119791.
Ferreira, J.J.M., Fernandes, C.I. and Ferreira, F.A.F. (2019), “To be or not to be digital, that is the question: Firm innovation and performance”, Journal of Business Research, Elsevier, Vol. 101 No. November 2018, pp. 583–590.
Fitzgerald, M., Kruschwitz, N., Bonnet, D. and Welch, M. (2013), “Embracing Digital Technology: A New Strategic Imperative”, MITSloan Management Review, pp. 1–12.
Gupta, G. and Bose, I. (2018), “Strategic learning for digital market pioneering: Examining the transformation of Wishberry’s crowdfunding model”, Technological Forecasting and Social Change, Elsevier, No. December 2017, pp. 1–12.
Nambisan, S. (2017), “Digital entrepreneurship: Toward a digital technology perspective of entrepreneurship”, Entrepreneurship Theory and Practice, Vol. 41 No. 6, pp. 1029–1055.
Presch, G., Dal Mas, F., Piccolo, D., Sinik, M., & Cobianchi, L. (2020). The World Health Innovation Summit (WHIS) platform for sustainable development. From the digital economy to knowledge in the healthcare sector. In Patricia Ordonez de Pablos & L. Edvinsson (Eds.), Intellectual Capital in the Digital Economy. London: Routledge, pp 19-28 doi 10.4324/9780429285882-4
You should also consider adding some references from the journal "Sustainability," to show that your paper fits the current dialogue of the journal.
There are several typos to solve (see for instance in the abstract: This current study was conductedin order to develop.... - change conductedin). Please check them.
Author Response
Dear Reviewer,
Thank you very much for your valuable comments and suggestions to improve our paper! We have tried to address to all your suggestions. We have marked all those changes in green color. We attach the Response Letter with the details of the required changes.
Thank you very much once again!
Best regards!

Reviewer 3 Report
The article meets the requirements for scientific research. This applies to the substantive and structural layer. The subject of the article is original, interesting and still relevant. In principle, I have no comments on the content and the editorial layer of the study.
Author Response
Dear Reviewer,
We highly appreciate your valuable comments and appreciations for our work! We thank you very much for this! We have performed improvments into the paper according to all the reviewers'suggestions. They are marked in yellow and green color, respectively.
Thank you very much once again!
Best regards
Round 2
Reviewer 1 Report
The answers to reviewers comments and introduced changes are satisfactory and in my opinion the authors improved the paper.